# Gene Self-Expressive Networks as a Generalization-Aware Tool to Model Gene Regulatory Networks

**DOI:** 10.3390/biom13030526

**Published:** 2023-03-13

**Authors:** Sergio Peignier, Federica Calevro

**Affiliations:** 1INSA Lyon, INRAE, BF2I, UMR 203, Université de Lyon, 69100 Villeurbanne, France; 2INRAE, INSA Lyon, BF2I, UMR 203, Université de Lyon, 69100 Villeurbanne, France; federica.calevro@insa-lyon.fr

**Keywords:** self-expressiveness, gene regulatory networks, regression, regularization

## Abstract

Self-expressiveness is a mathematical property that aims at characterizing the relationship between instances in a dataset. This property has been applied widely and successfully in computer-vision tasks, time-series analysis, and to infer underlying network structures in domains including protein signaling interactions and social-networks activity. Nevertheless, despite its potential, self-expressiveness has not been explicitly used to infer gene networks. In this article, we present Generalizable Gene Self-Expressive Networks, a new, interpretable, and generalization-aware formalism to model gene networks, and we propose two methods: GXN•EN and GXN•OMP, based respectively on ElasticNet and OMP (Orthogonal Matching Pursuit), to infer and assess Generalizable Gene Self-Expressive Networks. We evaluate these methods on four Microarray datasets from the DREAM5 benchmark, using both internal and external metrics. The results obtained by both methods are comparable to those obtained by state-of-the-art tools, but are fast to train and exhibit high levels of sparsity, which make them easier to interpret. Moreover we applied these methods to three complex datasets containing RNA-seq informations from different mammalian tissues/cell-types. Lastly, we applied our methodology to compare a normal vs. a disease condition (Alzheimer), which allowed us to detect differential expression of genes’ sub-networks between these two biological conditions. Globally, the gene networks obtained exhibit a sparse and modular structure, with inner communities of genes presenting statistically significant over/under-expression on specific cell types, as well as significant enrichment for some anatomical GO terms, suggesting that such communities may also drive important functional roles.

## 1. Introduction

### 1.1. Gene Regulatory Network Inference

Important phenomena in biology, such as morphogenesis, development, cell differentiation, cell-death and adaptation of organisms to changing conditions are governed, to an important extent, by the regulation of the expression of each gene, by a subset of specialized regulatory modules [1]. Such complex regulatory relationships are modeled as oriented graphs, termed Gene Regulatory Networks (GRNs), where nodes represent all genes, and edges represent regulatory links between Transcription Factors (TFs) and their Target Genes (TGs) [2]. In order to deepen the current knowledge on complex mechanisms shaping living organisms, different algorithms that study gene expression matrices to reverse-engineer GRNs have been proposed in the systems biology literature [2]. These algorithms have been classified in four major families [2]:

#### 1.1.1. Multi-Network Methods

These techniques aim at inferring GRNs by considering heterogeneous sources of data simultaneously [2]. Indeed, besides using gene expression data, these methods also rely on TF binding site patterns, or Chromatin Immuno-Precipitation data (e.g., [3,4]).

#### 1.1.2. Probabilistic Model-Based Methods

These techniques aim at inferring GRNs by fitting the parameters of a pre-established probabilistic model (e.g., Gaussian graphical models, Bayesian networks), with respect to experimental data [2]. Some models, termed Probabilistic Models, are grounded in probability theory, and they include approaches such as Bayesian networks and Gaussian Graphical Models (e.g., [5]).

#### 1.1.3. Dynamical Model-Based Methods

These methods aim at modelling the temporal changes in the expression of genes, through the parametrization of a dynamical model (e.g., Boolean Networks, Dynamic Bayesian Networks and Ordinal Differential Equations). Interestingly, once calibrated, those models can be used to simulate and analyze the biological systems *in-silico*. The reader can refer to [6] for a review article on probabilistic and dynamical model-based GRN inference.

#### 1.1.4. Data-Driven Methods

These techniques aim at analyzing high-throughput datasets, to score the level of *dependency* between each regulator and each gene, and then select highly scored links to reconstruct a GRN [2]. These methods are popular due to their speed, their simplicity, and their accuracy [2]. In practice, different measures have been used to score regulatory links, giving birth to two major groups:(i)The first group of methods is based on the assumption that a gene and its regulators should exhibit correlated gene expressions, and uses *correlation* (e.g., [7]) or more sophisticated information theory statistics such as *Mutual Information*, to score regulatory links (e.g., [8]).(ii)The second group of methods relies on the assumption that it should be possible to predict the expression of a target gene from the expressions of its regulators, and aim at training algorithms to model the expressions of each gene from those of the regulators. According to [2], this paradigm is among the most popular due to its scalability, and its ability to capture gene expression’s high-order conditional dependencies, that could not be captured by correlation or mutual information-based methods. In this second family of methods, a *feature importance score* is assigned to each regulator depending on its importance in the prediction task. Finally a subset of regulatory links is often chosen to define a putative GRN, by selecting for instance the *k* links with the highest scores. In practice, mainly regression algorithms have been used to this aim (e.g., [9,10]) but recently some works have also used classification algorithms (e.g., [11]). Unlike traditional machine learning applications, most GRN inference methods based on classifiers and regressors do not split the data into *training* and *test* subsets, and tend to train the models and compute the feature importance scores relatively to the *same dataset*, and thus such scores may not reflect the generalization capabilities of the predictive features, inducing potential misleading interpretations.

### 1.2. Generalization

According to [12], *generalization* is a major goal of Machine Learning algorithms, that aims at predicting accurately the outputs of examples that were not used during the training phase. Generalization errors are directly linked to the model’s level of underfitting or overfitting. According to [13] and [14], underfitted models lack of expressiveness, and tend to ignore important explanatory variables, leading to biased estimations and predictions. While overfitted models, contain more parameters than needed, and tend to include some training set residual noise in their underlying structures, incorporating for instance misleading variables. Overfitting risk is particularly high when the data includes many variables that are not related to the predicted variable or when there are few training examples, incurring in the so-called Freedman’s paradox [15]. Overfitted models exhibit a degraded generalization: they perform well on the training data, and poorly on unseen data.

In practice, both scenarios should be avoided, and many techniques have been proposed so far to balance the trade-off between underfitting and overfitting [14]. Some methods aim at penalizing complex models applying regularization (e.g., [16]), Bayesian priors (e.g., [17]), or model-specific techniques such as early algorithm stopping (e.g., [18]) and dropout in artificial neural networks (e.g., [19]) or pruning in tree-based algorithms (e.g., [20]). Other methods, such as the well-known cross-validation technique (e.g., [21]), aim at estimating the models generalization ability on unseen data. Estimating properly the underfitting/overfitting trade-off is particularly important in the context of GRN inference. Indeed, even if a good generalization capability does not guarantee that the model has captured the full network structure, assessing this skill can be used to filter out links provided by unadapated models: A link highly scored by a poorly generalizable model should not be retained at expenses of a link with a lower score assigned by a highly generalizable model. Previous approaches on GRN inference have focused on methods to avoid overfitting, using regularization (e.g., [10,22]), bayesian priors (e.g., [23]), or ensemble learning (e.g., [9,23,24]). Nevertheless, previous GRN inference works do not consider the estimation of the regressor’s generalization, neither to reverse-engineer GRNs, tune parameters, or evaluate the methods.

### 1.3. Self-Expressiveness Applications

*Self-expressiveness* is a mathematical property that has been used to characterize the relationship between instances in a dataset: it states that a data point laying in a union of subspaces can be expressed as a linear combination of other data points from the same subspace [25]. This property was introduced and applied in the domain of Subspace Clustering [25], an important unsupervised learning technique that has been extensively applied to computer vision tasks, such as image compression [26], image segmentation [27], video segmentation [28] or motion segmentation [29], and has also been extended to time series analysis [30]. The first, and most important step in subspace clustering aims at characterizing the relationship between pairs of data points, by applying the self-expressiveness property to learn a so-called self-expressive affinity matrix. Then, a second step aims at applying spectral clustering to the affinity matrix in order to cluster data points [31]. Interestingly, the self-expressiveness property has also been used to formalize the inference of Directed Acyclic Graphs (DAG) as an sparse linear self-expressive structure learning problem [32]. Recent extensions have proposed the use of Graph Neural Networks [33], and Reinforcement Learning [34] to solve an analogous non-linear alternative for the self-expressive structure learning problem. In practice, these techniques have been used to infer protein signaling networks from proteins and phospholipids expression levels [34] and twitter activity causal network [35], nevertheless self-expressiveness property has not been explicitly used in the context of GRN inference.

### 1.4. Contribution of the Paper

In this work we introduce Generalizable Gene Self-Expressive Networks, a new, simple, interpretable, and predictive formalism to model gene networks. We describe the relationship between this new formalism and previous regression-based GRN inference methods. We present two methods, termed GXN•EN and GXN•OMP, based respectively on ElasticNet [36] and OMP [37], that aim at inferring, assessing and tuning Generalizable Gene Self-Expressive Networks. Moreover, we evaluate the generalization capabilities of new approaches and state-of-the-art methods, introducing a new internal measure to assess such algorithms, showing the importance of developing generalization-aware methods. The evaluation of these new methods was performed using (i) Four Microarray datasets from the well-known DREAM5 benchmark [24], including three datasets from real organisms, namely *Escherichia coli*, *Streptococcus aureus* and *Saccharomyces cerevisiae*, and an in silico simulated dataset, (ii) Three RNA-seq datasets from multiple tissues of complex eukaryotes, namely *Canis familiaris*, *Rattus norvegicus* and *Homo sapiens*. Finally, in order to illustrate the potential of this methodology to perform differential regulatory communities analysis, we applied the GXN•OMP method on a control/disease RNA-seq case study, that reports the levels of expression of *H. sapiens* genes in different parts of the brain of Alzheimer disease patient and control donors.

For the sake of reproducibility, the experiments and the methods implementations are available online in a dedicated gitlab repository https://gitlab.com/bf2i/gxn (last updated on 20 February 2023) and the software can be installed as a Python library, from the Python Package Index (PyPi), the official third-party software repository for Python https://pypi.org/project/GXN/ (last updated on 20 February 2023).

## 2. Material and Methods

### 2.1. Self-Expressiveness Property

According to the *self-expressiveness* property, a data point that lays in a union of subspaces, can be expressed as a linear combination of other datapoints from the same subspace [25]. More formally, let $X\in R^{D\times N}$ be a dataset with *N* instances $X_{🟉,j}\in R^{D},\;\forall j\in \left\{1,\ldots ,N\right\}$. Data points are described in *n* unknown subspaces $\left\{S_{1},\ldots ,S_{n}\right\}$, such that each subspace $S_{k}\subset R^{D}$ has an unknown number $d_{k}$ of dimensions, and ${⋃}_{k=1}^{N}S_{k}\subseteq R^{D}$. Moreover, let $C\in R^{N\times N}$ be a coefficients matrix, such that $C_{i,j}$ denotes the element at row *i* and column *j*. The *self-expressiveness* property can be stated as follows:(1)$$\begin{array}{cc} \; & X_{🟉,j}=X\cdot C_{🟉,j}\;s.t.\;C_{j,j}=0\;\forall \;j\in \left\{1,\ldots ,N\right\}\\ \Leftrightarrow \; & X=X\cdot C\;s.t.\;C_{j,j}=0\;\forall \;j\in \left\{1,\ldots ,N\right\} \end{array}$$

Among the possible solutions that Equation (Equation 1) may have, the literature on self-expressiveness focuses on so-called *subspace preserving* solutions, that expresses data points as linear combinations of other points laying in the same subspace, i.e., matrix *C* where $C_{i,j}\neq 0$ when $X_{i}$ and $X_{j}$ lay in the same subspace, and $C_{i,j}=0$ otherwise.

In order to find a *subspace preserving* matrix *C*, the methods proposed so far (e.g., [25,38]) consist in regularizing *C* with a given norm denoted as ||C||, through the optimization of the objective function defined in Equation (Equation 2).
(2)$$\begin{array}{cc} \; & C_{🟉,j}^{*}=\underset{C_{🟉,j}}{\mathrm{argmin}}\;∥C_{🟉,j}∥\\ \mathrm{with}\;X_{🟉,j}=X\cdot C_{🟉,j}\; & \mathrm{and}\;C_{j,j}=0\;\forall \;j\in \left\{1,\ldots ,N\right\} \end{array}$$

Previous works relied on sparsity-inducing norms to remove connections between points laying in different subspaces, to ensure the *subspace preserving* property. Sparsity-inducing norms used so far include: $\ell_{1}$ regularization based on LASSO (e.g., [25,31]), $\ell_{0}$ regularization based on Orthogonal Matching Pursuit (e.g., [38,39]), and nuclear norm by means of a Low-Rank Representation [40]. Previous works [31] have also relied on the ElasticNet regularization, including both $\ell_{1}$ and $\ell_{2}$ norms, in order to stabilize the learning task and enable some connectivity between data points from different subspaces, ensuring the so-called *connectedness* property [31].

### 2.2. Generalizable Gene Self-Expressive Networks Inference

In this article, we aim at inferring Generalizable Gene Self-Expressive Networks (GXN) from gene expression matrices, derived from high-throughput RNA-seq or Microarray experiments, that quantify the expression of genes in different experimental conditions [2]. More formally, let $X\in R^{D\times N}$ denote a gene expression matrix, such that *D* is the number of conditions, while N=|Γ| denote the number of genes considered in *X*, where Γ is the set of all genes. Moreover let $X_{c,g}$ be the level of expression of geneg in conditionc, let $X_{🟉,g}$ be the vector of expression of a gene g∈Γ, across all conditions. In order to guide the inference task, one may also consider Ψ⊆Γ a subset of genes that can control the expression of other genes. The submatrix containing the levels of expression of such regulators, across all conditions, is denoted as $X_{🟉,\Psi }$. In practice, Ψ can represent the set of Transcription Factors, but if such a subset is not available, all genes could be considered as potential regulators, i.e., Ψ=Γ.

The GXN inference task can be formulated as an objective function analogous to Equation (Equation 2), that aims at finding a regularized *subspace preserving* matrix *C* for a given gene expression dataset *X*. Whenever a subset of regulators Ψ is available, a supplementary constrain should be incorporated, in order to ensure that only regulators are used as predictive features, i.e., $\forall \left(j,g\right)\in \Gamma {}^{2}$, such that ∀j∉Ψ, then $C_{j,g}=0$. In this context, the constrain stated in Equation (Equation 2), ensuring that $C_{g,g}=0\;\forall g\in \Gamma$, implies that the expression level of each regulator should be expressed as a linear function of other regulators (distinct from itself). This constrain is particularly important, since removing it would lead to a degenerate behaviour, where each regulator would be regulated only by itself, which would prevent the system to infer interactions between regulators. In addition to the aforementioned constrains, further constrains could also be included in order to integrate information from external datasets, such as ATACseq, CHIPseq or Transcription Factor Binding Sites motifs.

In practice, we address this task by training, for each gene g∈Γ, a linear regressor with meta-parameters θ and coefficients $C_{🟉,g}$, to predict $X_{c,g}$ from the regulators’ expressions $X_{c,\Psi }$, for any conditionc. In order to consider the models generalization capabilities, we relied on a nested cross-validation procedure [41], in order to avoid overly-optimistic estimations of the generalization capabilities [42]. An inner cross-validation step is used to select suitable parameters for each regressor, that lead to higher generalization capabilities. Besides that, an outer cross-validation step is executed on conditions that were not used to train the model and tune the parameters. This second step allows to monitor the generalization capabilities of each regressor, and eventually set to zero the coefficients of regressors suffering from poor generalization capabilities.

More formally, the gene expression matrix *X* is first divided in $k_{outer}$ non-intersecting equal-sized subsets $X_{fold_{1},\Gamma },X_{fold_{2},\Gamma },\ldots X_{fold_{k_{outer}},\Gamma }$, such that $\forall i\in \left\{1,\ldots ,k_{outer}\right\}\;X_{fold_{i},\Gamma }$ describes the expression levels of all genes Γ in $D/k_{outer}$ conditions. Then, the outer cross-validation defines iterativelly each subset $X_{fold_{i},\Gamma }$ as test set to assess the generalization capabilities of the model, while the remaining $k_{outer}-1$ subsets are concatenated and used as a learning dataset $X_{learn,\Gamma }$ for model selection and training. The inner cross-validation sub-divides $X_{learn,\Gamma }$ in $k_{inner}$ non-intersecting equal-sized subsets $X_{fold_{1}^{l},\Gamma },X_{fold_{2}^{l},\Gamma },\ldots X_{fold_{k_{inner}}^{l},\Gamma }$. Similarly, the inner cross-validation takes each subset $X_{fold_{j}^{l},\Gamma }\;\forall j\in \left\{1,\ldots ,k_{inner}\right\}$ as validation set to select the best model, while the remaining subsets are concatenated and used to train models with different parameter settings, and the model with the parameters that exhibited the best average performance on the validation datasets is selected, and retrained on the entire learning dataset. In this work, the number of folds for the inner and the outer cross-validations where set to $k_{inner}=5$ and $k_{outer}=5$ respectively.

In this work, we propose two Generalizable Gene Self-Expressive Network inference methods, called respectively GXN•OMP and GXN•EN.

### 2.3. GXN•OMP

The first method proposed in this paper relies in the well-known Orthogonal Matching Pursuit algorithm [37], that aims at solving a linear regression task subject to a sparsity constrain ensuring that only $d_{0}$ nonzero coefficients are used. More formally, GXN•OMP aims at solving the objective function stated in Equation (Equation 3).
(3)$$\begin{array}{cc} \; & C_{🟉,g}^{*}=\underset{C_{🟉,g}}{\mathrm{argmin}}\;{∥X_{🟉,g}-X\cdot C_{🟉,g}∥}_{2}^{2}\\ \mathrm{with}\;∥C_{🟉,g}{∥}_{0}\leq d_{0}, & \;\;C_{g,g}=0\;\;\forall g\in \left\{1,\ldots ,N\right\}\mathrm{and}\;C_{j,g}=0\;\;\forall j\notin \Psi \end{array}$$

To solve this task, OMP relies on a greedy forward feature selection method. At each step, the method selects the feature with the highest correlation with the current residual, then it updates the regression coefficients and recomputes the residual using an orthogonal projection on the subspace of the previously selected features. Moreover, an inner cross-validation step is used to select the parameter $d_{0}$ in a range between 0 and the hyper-parameter $d_{0}^{max}$ defining the maximal number of features. In practice, hyper-parameter $d_{0}^{max}$ can be set as a fraction δ of the number of regulators |Ψ|, or as the rank of matrix $X_{🟉,\Psi }$, whenever this values is lower, $d_{0}^{max}=min(\delta \times |\Psi |,\;rank\left(X_{🟉,\Psi }\right))$.

### 2.4. GXN•EN Algorithm

The second method proposed in this paper relies in the well-known ElasticNet regression technique [36], that addresses the linear regression task using simultaneously $\ell_{1}$ and $\ell_{2}$ regularization. More formally, GXN•EN addresses the objective function stated in Equation (Equation 4).
(4)$$\begin{array}{cc} \; & C_{🟉,g}^{*}=\underset{C_{🟉,g}}{\mathrm{argmin}}\;\frac{∥X_{🟉,g}-X\cdot C_{🟉,g}{∥}_{2}^{2}}{2D}+\alpha \rho ∥C_{🟉,g}{∥}_{1}+\frac{\alpha (1-\rho )}{2}{∥C_{🟉,g}∥}_{2}^{2}\\ \; & \mathrm{with}\;C_{g,g}=0\;\forall \;g\in \left\{1,\ldots ,N\right\}\;\mathrm{and}\;C_{j,g}=0\;\;\forall \;j\notin \Psi \end{array}$$

According to [36], pure $\ell_{2}$ regularization (Ridge regression) shrinks the coefficients of correlated features while keeping them non-zero, while pure $\ell_{1}$ regularization (Lasso regression) tends to select only one feature among a set of correlated dimensions, and exhibits a degenerate behaviour on highly correlated datasets. ElasticNet is a compromise between Ridge and Lasso regression, and the trade-off between $\ell_{1}$ and $\ell_{2}$ penalties is governed by a mixing parameter ρ∈[0,1], when ρ=0 only an $\ell_{2}$ regularization is used, and the task corresponds to a Ridge Regression, while for ρ=1 only a $\ell_{1}$ regularization is used, and the problem is equivalent to a Lasso Regression. For 0<ρ<1 the regularization is in between Lasso and Ridge. The overall regularization strength is governed by a parameter α, when α=0 the task is equivalent to an ordinary least squares regression, and higher values of α increase the impact both of the $\ell_{1}$ and $\ell_{2}$ penalties simultaneously. For a given value of α, the sparsity of the solutions are enforced by higher values of ρ, and according to [36], when ρ is close to 1, it tends to select few nonzero coefficients as Lasso, while avoiding degenerate behaviour on correlated data.

In practice, both parameters where chosen using the inner cross-validation technique previously introduced, exploring a grid of parameters α and ρ. Since we focus on *subspace preserving* solutions, that express each gene expression as a linear combination of the expressions of the regulator genes that lay in the same subspace, rather large values for parameter ρ should be preferred, in order to enforce a higher sparsity, in this work we specifically explored the range ρ∈{0.8,0.9,0.99,1}. Regarding parameter α, we used the so-called pathwise coordinate descent method [36] to explore solutions for a decreasing sequence of $K_{\alpha }$ values α drawn on a log scale from $\alpha_{max}=\frac{max_{i\neq j}\left(\right|X_{🟉,i}^{⊺}\cdot X_{🟉,j}\left|\right)}{D\rho }$ (for which the coefficients vector is null) and a value $\alpha_{min}=\epsilon \alpha_{max}$, with 0<ϵ<1. In this work we relied on a random coefficient update technique, that modifies a random coefficient at every iteration of the optimization method rather than looping over features sequentially, since this technique tends to lead to faster convergence. In order to reduce the number of meta-parameters, in this work we set $\epsilon =\frac{1}{K_{\alpha }}$, intuitively the smaller the range between $\alpha_{max}$ and $\alpha_{min}$ the less values α need to be explored.

### 2.5. Relationship with GRN Inference

Even if the *self-expressiveness* and the *subspace preserving* properties have not been explicitly mentioned in the context of GRN reverse-engineering, GRN inference methods based on feature selection on regression algorithms rely on related principles. Indeed, the central assumption of this paradigm is related to the self-expressiveness property, since it states that the expression $X_{c,g}$ of each geneg for any conditionc, can be expressed as a function of the levels of expression $X_{c,\Psi }$ of regulator genes Ψ, for the same condition *c*. Regressors are trained by finding a set of parameters that minimize some kind of average regression error, on the training dataset. In practice, existing methods rely mainly on linear regressors (e.g., [10]) or regression trees (e.g., [9,43]) as base-learners. And they tend to train ensembles of regressors on randomized versions of the dataset (e.g., [10]), or on subsets/subspaces of the dataset (e.g., [9,43]). Then, for each geneg, a feature importance vector $\gamma^{\left(g\right)}\in R^{\left|\Psi \right|}$ is computed, such that $\gamma_{\psi }^{\left(g\right)}$ denotes the average *importance* of regulator ψ∈Ψ in the prediction of $X_{🟉,g}$. This measure is assumed to quantify the *dependency* between genes *g* and ψ. Notice that in this case, regressors are only used to score the links, and they do not model the genes expression quantitatively, as in the model-based family or in the formalism presented in this article. An important difference between the GXN formalism and existing regression-based GRN inference tools, is that the GXN formalism focuses on the predictive power of the generated models.

In the context of regression-based GRN inference, feature importance scores are often non-zero, and then a subset of the regulatory links requires to be selected to define an oriented graph G=〈Γ,E〉 as putative GRN, with the set of genes Γ as the set of nodes, and by selecting the *k* links with the highest scores as edges $E=\{\left(\psi ,g\right)\in \Psi \times \Gamma \;s.t.\;|\left\{\left(\psi^{\prime },g^{\prime }\right)\in \Psi \times \Gamma \;s.t.\;\gamma_{\psi^{\prime }}^{\left(g^{\prime }\right)}\geq \gamma_{\psi }^{\left(g\right)}\right\}|\leq k\}$. The size of the inferred GRN is defined by *k*, which is a capital parameter. However, without external information, choosing a suitable value for *k* may not be a trivial task.

Regarding Generalizable Gene Self-Expressive Networks, the subspace preserving matrix *C* can be considered as the adjacency matrix of an oriented graph G=〈Γ,E〉, with all genes Γ as vertices, and a set of links $E=\{\left(\psi ,g\right)\in \Psi \times \Gamma \;|\;C_{g,\psi }\neq 0\}$. Since the regularization applied to matrix *C* aims at providing sparse models, it is not necessary to consider an extra parameter *k* to select a subset of links in the case of the GXN formalism. According to the *subspace preserving* property, most connections between points laying in different subspaces are removed by sparsity-inducing regularization, and any link (ψ,g)∈E should connect a regressor ψ and a gene *g* if they lay in the same expression subspace. The underlying hypothesis that allows us to interpret such a GXN network as a GRN is that regulators that lay in the same expression subspace of a target gene should be involved in the control of its expression.

### 2.6. Datasets Description

#### 2.6.1. DREAM5 Dataset

In order to assess GXN•OMP and GXN•EN, and compare them to state-of-the-art approaches, we used the DREAM5 benchmark data [24]. This benchmark has three datasets from real organisms, namely *E. coli*, *S. aureus* and *S. cerevisiae*, and an in silico simulated dataset. Each dataset contains a gene expression matrix, a list of TFs, and a list of known regulatory links between TFs and their TGs, i.e., a gold standard GRN.

The *E. coli*, *S. aureus* and *S. cerevisiae* gene expression matrices are Microarray datasets, downloaded from Gene Expression Omnibus platform (GEO) (http://www.ncbi.nlm.nih.gov/geo, accessed on 6 February 2023). The authors of [24], applied to these datasets a Robust Multichip Averaging background adjustment, a quantile normalization, a probeset median polishing and a logarithmic transformation. Moreover, the lists of TFs for *E. coli*, *S. aureus* and *S. cerevisiae*, were determined by means of a Gene Ontology (GO) annotation conducted in [24], completed with manually curated TFs list from the RegulonDB 6.8 database [44] for *E. coli*, and a list of TFs provided in [45] for *S. cerevisiae*.

The gold standard *E. coli* GRN, includes only regulatory links with strong experimental evidence, from the RegulonDB 6.8 database [44]. The gold standard *S. cerevisiae* GRN, includes regulatory interactions determined in [46], through the study of ChIP-chip datasets and a TF binding sites motifs analysis. Regarding *S. aureus*, no experimentally validated GRN was available, nevertheless the set of prokaryotic regulatory interactions reported in the RegPrecise database [47], was used as a proxy of a GRN. Therefore, for this organism the quality of the GRN is lower than those of the other DREAM5 benchmark datasets, and it was included in [24] as a complementary way to asses the methods’ robustness.

The in silico dataset, was generated using the GeneNetWeaver software [48]. According to [24], the in silico GRN is a randomized version of the *E. coli* GRN, incorporating 10% of new random regulatory links. This GRN, was used to generate a gene expression matrix, using a dynamical system of Ordinary Differential Equations: GeneNetWeaver aims at modeling the dynamics of the concentration of mRNAs and proteins stored as two distinct numerical vectors. The concentration of each protein is determined linearly by the concentration of its corresponding mRNA and its translation rate. The concentration of regulatory proteins determine the rate of transcription of target genes into mRNA, through a gene network additive and multiplicative interaction model based on the thermodynamic assumption that each promoter of a target gene is occupied with a probability depending on the concentration of its corresponding regulatory protein and its dissociation constant, via a Hill equation. Both mRNAs and proteins are subject to linear degradation. Finally, the GeneNetWeaver software simulates perturbations in the network via i) gaussian perturbation, i) knockdown and ii) knock-out in silico experiments, and relies on the Runge-Kutta 4–5 solver to simulate the evolution of the GRN dynamical system. We refer the readers to [49] for a detailed description of the GeneNetWeaver software and its underlying model.

#### 2.6.2. RNA-seq Multi-Tissue Eukaryote Datasets

Given the lack of gold-standard datasets from complex eukaryotic organisms for GRN inference assessment, we have collected three RNA-seq datasets from eukaryotic organisms, namely *C. familiaris*, *R. norvegicus*, and *H. sapiens*, to evaluate GXN•OMP and GXN•EN methods in more challenging scenarios. Each dataset contains a RNA-seq gene expression matrix, a Gene Ontologies annotations database and a list of regulators (including TFs and cofactors). The Gene Ontology annotations for each dataset were downloaded from the Ensembl database [50], and the lists of regulators were downloaded from the AnimalTFDB database [51]. Regarding gene expression datasets, we downloaded the iDog *C. familiaris* database [52], this dataset contains a total of 75 conditions from 6 cell-types, namely 25 samples from MDCK cells, 4 from adrenal cortex tissue, 12 from heart tissue, 21 lymphoma cells, 9 from neuroretina cells and 4 from pituitary tissue. In addition, we downloaded the rat BodyMap database [53], which contains 80 conditions from 11 tissues, namely 8 samples from the adrenal gland, 8 from the brain, 8 from the heart, 8 from the kidney, 8 from the liver, 8 from the lung, 8 from the spleen, 8 from the thymus, 8 from the gastrocnemius, 4 from the uterus, and 4 from the testis. Finally, regarding the *H. sapiens* dataset, we downloaded and concatenated three datasets from different cell-types, namely the Macrophage Immune response [54] which contains 89 samples, the iPSC-derived Sensory Neurons [55] which contains 106 samples, and the 465 lymphoblastoid cell-lines [56] which contains 462 samples. The four last datasets were downloaded from the Gene Expression repository [57]. For these three organisms, and particularly for *C. familiaris* and *R. norvegicus*, the number of samples per tissue is not sufficient to run tissue-specific network inferences. Therefore, in this work we decided to combine samples from different tissues, running a single inference per organism, in order to have a larger sample with more variability. Unlike in the DREAM5 dataset, gold standard GRNs were not available for the RNA-seq eukaryotic datasets. In this work, we decided to focus on the interactions between regulator genes only, and thus any gene that is not a TF or a cofactor, was not included in the analysis. More formally, let Ψ be the set of genes encoding TFs or cofactors, and let Γ be the total set of genes considered in the analysis, then for these datasets we set Γ=Ψ. The aforementioned datasets were carefully analyzed as described in Appendix A, and no important batch effects could be detected, as depicted in Figure A1a–d. Therefore, it was not necessary to perform batch effect correction in this work. Nevertheless, depending on the particularities of other datasets, it could be necessary to carefully study whether a batch effect correction should be applied [58].

#### 2.6.3. RNA-seq Disease/Control Case Study

In order to illustrate how the GXN methodology can be used to perform differential regulatory communities analysis, we applied the GXN methodology on a control/disease RNA-seq case study. In practice, we analyzed the publicly available “Allen Brain Institute Aging, Dementia and TBI study” dataset [59]. This dataset contains batch-corrected RNA-seq gene expression matrices, reporting the level of expression of *H. sapiens* genes in 377 samples from four brain regions (i.e., temporal neocortex, white matter of forebrain, hippocampus and parietal neocortex), of 107 male and female elderly individuals from the Change in Thought (ACT) cohort [60]. According to the Diagnostic and Statistical Manual of Mental Disorders (DSM) clinical diagnosis, available for each patient, most samples belong to two categories, namely *No Dementia* (i.e., control) donors (193 samples) and *Alzheimer’s disease type* patients (110 samples). In order to focus on the most promising genes, we have filtered out genes that are not expressed in all samples. As described in the previous paragraph, we used the Ensembl database [50] *H. sapiens* The Gene Ontology annotations, as well as the AnimalTFDB database [51] lists of regulators for *H. sapiens*.

### 2.7. Evaluation against DREAM5 Gold Standard

The evaluation of the methods proposed in this article, against DREAM5 gold standards, was conducted as a binary classification task, where possible regulatory links are classified as true or false interactions, as in [24]. Regulatory links between TFs and TGs that were found experimentally, and reported in the benchmark datasets were considered as *true interactions* (regardless whether they are Up or Down regulatory links), while pairs of TFs and TGs for which the experimental study could not reveal a regulatory interaction were considered as *false interactions*. Then, depending on the GRN inference method outcome, four possible scenarios exist for each link: (i) True Positive: the method infers a true link, (ii) False Negative: the method fails inferring a true link, (iii) False Positive: the method infers a false link as being positive, and (iv) True Negative: the method does not predict a false link. According to [24], GRN gold standards only report the *experimentally tested subset* of all the true regulatory interactions, and thus to avoid penalizing methods for detecting true interactions remaining experimentally untested, links involving TFs or TGs that were not tested experimentally were excluded from the assessment [24].

More formally, let $\Gamma_{gold}^{}\subseteq \Gamma^{}$ and $\Psi_{gold}^{}\subseteq \Psi^{}$ be respectively the subset of genes and the subset of TFs that were experimentally studied, such that $\Psi_{gold}^{}\subseteq \Gamma_{gold}^{}$. Let a gold standard GRN be an oriented graph $G_{gold}^{}=\langle \Gamma_{gold}^{},E_{gold}^{}\rangle$, where $E_{gold}^{}\subseteq E_{gold}^{full}$ is the set of *true regulatory links* among the possible links between studied genes (excluding self loops) $E_{gold}^{full}=\left\{\left(\psi ,g\right)\in \Psi_{gold}^{}\times \Gamma_{gold}^{}\;|\;\psi \neq g\right\}$. Links in $E_{gold}^{full}\backslash E_{gold}^{}$ are taken as *false regulatory links*, while links in $\left(\Psi^{}\times \Gamma^{}\right)\backslash E_{gold}^{full}$ are not considered in the evaluation. The fraction of true regulatory links $|E_{gold}^{}\left|/\right|E_{gold}^{full}|$, shows that DREAM5 datasets exhibit a class imbalance, as reported in Table 1.

Finally, the evaluation [24] was achieved by means of standard evaluation measures for binary classification: the Area Under the Receiver Operating Characteristic curve (AUROC) [61], and the Area Under the Precision Recall curve (AUPR) [62] values.

### 2.8. Inner Evaluation Procedure

As described in the previous section, when experimentally validated regulatory links are available, evaluation measures for binary classification (e.g., AUROC, AUPR) should be used to evaluate GRN inference tools, then, we used this procedure for the DREAM5 datasets. Nevertheless, in order to assess inference methods on datasets lacking of gold stantard GRNs (i.e., *C. familiaris*, *R. norvegicus*, *H. sapiens* datasets), we have decided to consider the following inner evaluation metrics.

#### 2.8.1. Regressors Performance Measures

We assessed each TG regressor’s generalization skills by computing the well-known $R^{2}$
*determination coefficient* [63] with a cross-validation methodology. The $R^{2}$ coefficient represents the fraction of the variance, of a vector of observations, that is explained by the model. Applied to unseen data, this coefficient assesses the generalization and predictive capabilities of the regressor. This score approaches 1 when the regressor tends to explain perfectly the data, it is equal to zero when the predictive capabilities of the model are equivalent to a constant model predicting the data mean, and it can be arbitrarily negative in cases of unadapted models.

In addition, we have decided to compute the *runtime* that each algorithm takes to train the gene expression regressor models. In practice, we computed both measures for each cross-validation fold, and then reported average results for each regressor.

#### 2.8.2. Network Topology Assessment

We also decided to inspect the topology of the inferred GXN networks, by measuring the *sparsity* of the networks, as well as the distribution of the in-degrees and out-degrees of their constitutive nodes. More formally, let a directed graph G=〈Γ,E〉, with all genes Γ as vertices, and a set of links $E=\{\left(\psi ,g\right)\in \Psi \times \Gamma \;|\;C_{g,\psi }\neq 0\}$, represent the GRN encoded by the GXN model with subspace-preserving matrix *C*, as defined in Section 2.2. Moreover let $E^{full}=\left\{\left(\psi ,g\right)\in \Psi \times \Gamma \;|\;\psi \neq g\right\}$ be the set of all possible edges between regulators and genes excluding self-loops. The level of *sparsity* of *G* is the percentage of the possible links that were not included in the model, i.e., $sparsity=\frac{|E^{full}\backslash E|}{|E^{full}|}\times 100$. Moreover, let $deg^{-}\left(g\right)$ be the in-degree of the node representing gene *g*. The in-degree of *g* is defined as the number of regulators involved in the regression model predicting the expression of gene *g*, i.e., $deg^{-}\left(g\right)=\left|\left\{\psi \in \Psi \;|\;\left(\psi ,g\right)\in E\right\}\right|$. Finally, let $deg^{+}\left(\psi \right)$ be the out-degree of the node representing a regulator gene ψ. This measure is defined as the number of target genes including ψ in their gene expression regression models, i.e., $deg^{+}\left(\psi \right)=\left|\left\{g\in \Gamma \;|\;\left(\psi ,g\right)\in E\right\}\right|$.

### 2.9. RNA-seq Eukaryote Datasets Community Analysis

#### 2.9.1. Community Detection

In order to deepen the analysis of the networks inferred on the RNA-seq Eukaryote dataset, and given the lack of gold standard GRNs available, we decided to assess whether the gene networks inferred tend to structure in communities (i.e., sub-networks of genes densely intra-connected by regulatory links) sharing common functional roles. To do so, we used the Clauset-Newman-Moore greedy modularity maximization technique [64], that aims at partitioning a graph into a set of sub-graphs termed communities that exhibit maximal generalized modularity *Q*, as defined in [65] and formalized in Equation (Equation 5).
(5)$$\begin{array}{cc} \; & Q=\frac{1}{\left|E\right|}\sum_{g\in \Gamma ,\psi \in \Psi }\left(C_{\psi ,g}-r\cdot \frac{deg^{+}\left(\psi \right)\cdot deg^{-}\left(g\right)}{\left|E\right|}\right)\cdot \zeta \left(\psi ,g\right) \end{array}$$
where ζ:Ψ,Γ→{0,1} is a function such that ζ(ψ,g)=1 if ψ and *g* are in the same community and ζ(ψ,g)=0 otherwise, and $r\in R_{+}^{*}$ denotes the so-called *resolution* parameter, that controls the trade-off between intra and inter-community edges. In practice, the algorithm tends to output few large communities when *r* is small, and many small communities for higher values of *r*. As stated in Equation (Equation 5), in this work, the modularity was computed by considering the regression coefficients as the edges’ weights, and it ranges between 0 for a poor modularity, and 1 for a perfect modularity. In practice, the Clauset-Newman-Moore greedy modularity maximization algorithm at first considers each node as an independent community, and then at each iteration it joins the pair of communities that most increases modularity, until a single community is created. Then, the communities that maximize the generalized modularity *Q* are output.

The resolution parameter *r*, was determined using the well-known elbow method: For different values of r∈{0.5,0.6,…,5}, we ran the Clauset-Newman-Moore algorithm in order to partition the GXN in different communities, and then we computed the average Sum of Square Errors (SSE) between the gene expressions and the mean expression of their corresponding communities. More formally, let $\Gamma^{\kappa }\subseteq \Gamma$ denote the set of genes from community κ, and let $\bar{X_{🟉,\Gamma^{\kappa }}}=\frac{1}{|\Gamma^{\kappa }|}\sum_{g\in \Gamma^{\kappa }}X_{🟉,g}$ be the average expression of the genes from community κ, finally $SSE=\sum_{\kappa }\sum_{g\in \Gamma^{\kappa }}\left|\right|X_{🟉,g}-\bar{X_{🟉,\Gamma^{\kappa }}}{\left|\right|}_{2}^{2}$. Finally we applied the Kneedle algorithm [66], to determine the elbow in plots representing the SSE for different values of *r*, as depicted Figure 1.

#### 2.9.2. GSEA and GO Enrichment Analysis

In this work we used a Gene Set Enrichment Analysis (GSEA) [67] in order to determine communities of genes that are collectively over-expressed or under-expressed, with statistical significance, in a particular tissue or cell-type. In practice, we used the *difference-of-classes* ranking method, we assessed the result’s significance running 10,000 *gene-set* permutations and only kept relationships with a *p*-value and a False-Discovery-Rate (FDR) both lower than 0.05. Finally we extracted statistically over-represented GO terms for each community of genes, using the GOATOOLS [68] Python library. In practice, we used a default significance cut-off *p*-value equal to 0.05, with a Benjamini Hochberg multiple test False Discovery Rate (FDR) correction, and we only kept the relationships with a FDR lower than 0.05 and, for the sake of simplicity, we focus specifically on the descendent of the *anatomical structure development* (i.e., GO:0048856), among the set of GO terms.

### 2.10. Parameter Setting and Implementation

In order to analyze, on the DREAM5 datase, the impact of a rather large range of parameter values, but focusing with more details on sparser solutions, we have run experiments with δ={0.01,0.03,0.05,0.1,0.2,0.3,0.5} for the OMP algorithm, and ϵ∈{0.02,0.025,0.033,0.05,0.1,0.2} for the ElasticNet algorithm.

Moreover, we illustrated the performances of GXN•OMP and GXN•EN, on the Eukaryotes RNA-seq datasets, with $d_{0}^{max}=30$ and ϵ=1/3 respectively. Both parameter settings tend to produce simple models, with few coefficients (and also rather lower $R^{2}$ scores). Therefore using such a parameter setting, allows to assess the GXN•OMP and GXN•EN models, on a worse case scenario, where sparser and simpler models are used to deal with complex eukaryotic organisms’ RNA-seq datasets. In order to study the communities within the GXN models we only retain the links from models exhibiting an $R^{2}$ determination score higher than 0.5 for the validation set (i.e., models that exhibit a higher enough generalization performance).

The methods proposed in this paper were implemented using the GReNaDIne [69] package, and the Scikit-Learn [70] ElasticNetCV and OrthogonalMatchinPuirsuitCV implentations of the ElasticNet and OMP algorithms. Moreover, the evaluations were run using the GReNaDIne [69] package. The software to run the inferences and the analysis presented in this article is available at https://gitlab.com/bf2i/gxn (last updated on 20 February 2023).

Important characteristics of DREAM5 and RNA-seq datasets studied in this article are summarized in Table 1.

All experiments were executed on a 2,9GHz Intel Core i9 CPU, running macOS Big Sur 11.2.3, with a 32 Go RAM capacity.

## 3. Results

### 3.1. DREAM5 Datasets

#### 3.1.1. Models Topology

The GXN•OMP and GXN•EN models, exhibit in general high sparsity levels (>65%), which depend both on the parameter settings and the dataset characteristics, as shown in Figure 2. Regarding the parameter settings impact, increasing δ in the GXN•OMP method and decreasing ϵ in the GXN•EN algorithm, allows to build more complex models, with lower sparsity (Figure 2a,d), accordingly genes exhibit higher in-degrees (Figure 2c,f), and regulators present higher out-degrees (Figure 2b,e). Regarding the impact of the dataset characteristics, in general real datasets, and specially *S. cerevisiae* and *E. coli* datasets, tend to lead to more complex models (lower sparsity), with genes exhibiting larger in-degrees and regulators with higher out-degrees, while models inferred for the in silico dataset are sparser. The *S. aureus* dataset exhibit also rather simple models, but since this dataset contains few regulators, the sparsity tends to be lower than those of more complex *S. cerevisiae* and *E. coli* models.

In order to compare the sparsity obtained by GXN based methods, to those obtained by state-of-the-art techniques, we have extended the definition stated in Section 2.8 to methods based on feature importance, by computing the sparsity as the percentage of feature importance scores that are close to zero. In practice, we consider scores lower than a threshold equal to $10^{-5}$. As reported in Table 2, state-of-the-art methods tend to exhibit low sparsity levels (<1%), unlike GXN based methods, as described in Section 2.5.

#### 3.1.2. Regressors Performance

In Figure 3, we represent, for the different algorithms, datasets, and different parameter settings, the average $R^{2}$ determination coefficient, and training run-time, measured throughout the outer cross-validation procedure for each gene expression regression task. In general, for both GXN•EN and GXN•OMP, when the number of coefficients increases, the $R^{2}$ coefficient increases at first, and then tends to reach a plateau. Indeed, GXN•EN models exhibit significantly better results when ϵ decreases from 0.2 to 0.05, and tends to plateau for ϵ≤0.033. Similarly, GXN•OMP models exhibit higher $R^{2}$ scores when δ increases from 0.01 to 0.1, and reaches a plateau for δ≥0.1. Moreover, the $R^{2}$ scores obtained for each method with the same parameter setting, depend to an important extent on the analyzed dataset. In practice, real world datasets lead to higher determination coefficients, while the results obtained for the in silico dataset tend to be poorer. Compared to the state-of-the-art results, the best GXN•EN models exhibit similar results to the Support Vector Machine Regressor (SVR) ones for the real datasets, and better results than those obtained by the Random Forest Regressor (RF) ones. While the best GXN•OMP models, despite being slightly worse than the SVR results, tend to exhibit comparable results with respect to the RF results, on the real datasets. Regarding the in silico dataset, both GXN•EN and GXN•OMP models exhibit lower determination coefficients than SVR and RF. Despite the high sparsity and simplicity of GXN•EN and GXN•OMP models, they exhibit mostly positive determination coefficients, with levels that are comparable with respect to more complex state-of-the-art tools. Run-times required to fit each regressor, represented in Figure 3, show that GXN•EN and GXN•OMP are rather fast methods, such as SVR, since they require in most cases between $10^{-3}$ and 1 second to fit each TG regressor, depending on the parameter setting and the dataset. Whereas the RF is a very time consuming method, that requires between $10^{-1}$ and 10 seconds to fit each TG regressor. Moreover GXN•OMP algorithm is faster than GXN•EN, since in most cases GXN•OMP fits each TG regressor in less than $10^{-1}$ seconds, while GXN•EN requires between $10^{-1}$ and 1 second in most cases.

#### 3.1.3. GRN External Evaluation

Figure 4 represents, for the different algorithms, and different parameter settings, the AUROC and AUPR evaluation scores on the DREAM5 datasets. In general, for both GXN•EN and GXN•OMP, increasing the models complexity tends to lead to slightly better scores, and then tends to reach a plateau. Indeed, the GXN•OMP models tend to exhibit slightly better AUROC and AUPR scores when δ increases, nevertheless this trend is small, and scores are mostly stable. Similarly, for the real dataset, GXN•EN GRNs exhibit higher, or at least equal AUROC and AUPR scores when ϵ decreases from 0.2 to 0.033 and then the trend tends to plateau for ϵ≤0.1, but the method exhibits an opposite trend, for the in silico dataset. Moreover, the best results obtained by the GXN•OMP and the GXN•EN methods are comparable with respect to those obtained by SVR and RF.

Indeed, the average AUROC score obtained, on the DREAM5 datasets, by GXN•EN (with ϵ=0.02), SVR and GXN•OMP (with δ=0.5) are very similar (respectively 0.725, 0.71 and 0.7), while RF exhibited a lower average AUROC=0.67. In terms of average AUPR, both GXN•EN and GXN•OMP obtained the highest average score AUPR=0.26, outperforming SVR and RF, which obtained respectively 0.235 and 0.22.

### 3.2. RNA-seq Multi-Tissue Eukaryotic Datasets

The networks inferred from the *C. familiaris*, *R. norvegicus* and *H. sapiens* multi-tissue RNA-seq datasets exhibit a high sparsity level (≥99%), as reported in Table 3. The median number of regulators involved in the control of each gene, determined the GXN•EN method, is close to 10 regulators for all datasets, as shown in Figure 5b. While, using the same parameter setting, the GXN•OMP determined a similar number of regulators for the *C. familiaris* and *R. norvegicus* datasets, but has built, for the *H. sapiens* dataset, models with twice more regulators in median, as shown in Figure 5b. This suggests that the GXN•OMP is more flexible and better adapts to the particularities of each dataset.

Moreover, these networks exhibit a modular structure, they comprise communities of genes that share more connections between them, than with genes from other communities. According to [64], in practical cases, a modularity above 0.3 is a significant indicator that the given network exhibits a strong community structure. The networks obtained in this work exhibit high modularities (≥0.5), as reported in Table 3. In general, the networks obtained using GXN•OMP tend to be splitted into more communities, which were also more homogeneous in size, while those obtained using GXN•EN tend to be splitted in fewer communities, and to be heterogeneous in size, as shown in Figure 5a. Moreover, the *R. norvegicus* and *C. familiaris* networks exhibited more communities than the *H. sapiens* datasets, which could possibly reflect the fact that the former networks were obtained from datasets containing more tissues than the *H. sapiens* datasets.

In terms of inner quality, both methods allow to obtain in general high validation $R^{2}$ determination coefficients for most TGs, nevertheless GXN•OMP allowed to obtain better results (median $R^{2}$ scores between 0.8 and 0.9) than the GXN•EN (median $R^{2}$ scores between 0.7 and 0.8) as shown in Figure 6a. Moreover, GXN•OMP models required between 2 and 5 times less run-time to be fitted than GXN•EN ones, as depicted in Figure 6b.

In summary, for the *C. familiaris*, *R. norvegicus* and *H. sapiens* RNA-seq datasets, GXN•OMP models outperform GXN•EN ones in terms of $R^{2}$ coefficients and run-times, both methods exhibit similar results in terms of sparsity, and GXN•OMP exhibits better modularities for *C. familiaris* and *R. norvegicus* but worse for *H. sapiens*.

Finally, the communities that were identified in the GXN•OMP and GXN•EN models were tested, through a GSEA, for enrichment in terms of expression on specific tissues, as well as for enrichment of anatomical GO terms. Several GXN•OMP and GXN•EN communities clearly exhibit significant GSEA Normalized Enrichment Scores (i.e., the genes from each community exhibit concordant, and statistically significant, over/under-expression on specific tissues), as well as statistically over-represented anatomical GO terms. Interestingly, some GXN•OMP and GXN•EN communities present even both significant GSEA and GO enrichment Scores exhibiting biological coherence, as depicted in Figure 7, Figure 8 and Figure 9, suggesting that such communities may have functional roles in the different tissues represented in the RNA-seq datasets.

For instance: (1) In the *C. familiaris* dataset, communities 0 from both GXN•OMP and GXN•EN, exhibit important enrichment scores for the heart development GO term. (2) In the same dataset, community 23 from GXN•OMP and community 4 from GXN•EN exhibit important enrichment scores for the neuroretina and for the retina development GO terms. (3) similarly, in the *R. norvegicus* dataset, communities 39 and 77 from GXN•OMP and community 5 from GXN•EN present an enrichment score for the brain, as well as GO terms related to the development of the nervous system. (4) Communities 5 and 6 from GXN•OMP are enriched in the uterus and also exhibit an uterus development enrichment GO term (5) Community 15 from GXN•EN is enriched in the liver and also exhibits GO terms related to the development of liver and pancreas. (6) In the *H. sapiens* dataset, Communities 8, 17, 20, 23 and 25 from GXN•OMP and 1 and 10 from GXN•EN are enriched for the neuron, and also present different GO terms relative to the development of the nervous system.

In order to deepen the analysis regarding intra-communities regulatory interactions, we represent, in Figure 10, the regulatory links of communities Cf|23 and Rn|39 (the full list of TFs from these communities are presented in Table A1). Interestingly, the community Cf|23 from the GXN•OMP model of the *C. familiaris* dataset, contains important regulators involved in the development of the neuroretina. For instance POU6F2 is involved in the development of retinal ganglion and amacrine cells [71], PRDM13 is involved in the development of amacrine cells [72] and IRX6 is a key regulator of retinal interneuron subtype identity. Regarding community Rn|39 from the GXN•OMP model of the *R. norvegicus* dataset, it contains important TFs involved in the development of neurvous system. For example Fezf1, a well conserved TF that controls neurogenesis and cell-fate in the mammalian nervous system development [73], Neurog2, a TF involved in determining neuronal type fate [74], and POU3F2, a TF plying a key role in brain development [75].

Interestingly, the confusion matrices between GXN•OMP and GXN•EN communities for the *C. familiaris*, *R. norvegicus* and *H. sapiens* datasets, shown Figure 11, exhibit a strong correspondence. A Pearson’s $\chi^{2}$ test of independence for each on the the matrices, reveals that the independence hypothesis between the GXN•OMP and GXN•EN communities is rejected with corresponding $\chi^{2}$ statistics of 14,233.87 for *R. norvegicus*, 9,803.57 for *H. sapiens* and 11,685.36 for *C. familiaris*, which correspond in all cases to near 0 *p*-values, suggesting that the communities structures obtained using both methods are robust.

### 3.3. Alzheimer/Control Patients Case Study

Comparing two groups of gene expression samples (e.g., control-disease, control-treatment), in order to identify genes that characterize each group is an important task in biomedical and biological applications. For instance, Differential Gene Expression analysis, is a classical methodology that aims at comparing gene expression levels between two groups of samples, to identify significantly up-regulated or down-regulated genes in one condition compared to the other one [76]. In this work, we illustrate how the GXN methodology can be used to identify significantly up-regulated or down-regulated communities within a GRN. To do so, we analyzed the Alzheimer/Control *H. sapiens* RNA-seq dataset, described in Section 2.6, using the GXN•OMP method with parameter $d_{0}^{max}=30$, since this technique performed better than GXN•EN, on all the other RNA-seq datasets. Then, we applied the different steps described in Section 2.9: (i) The Clauset-Newman-Moore technique [64] was used to partition the GXN into communities (with a resolution parameter r=1.75 determined as explained in Section 2.9), (ii) the GSEA [67] was run to determine communities of genes that are collectively over-expressed or under-expressed in Control or Alzheimer conditions, and (iii) the GOATOOLS [68] library was used to extract statistically over-represented GO terms (descendants of the *biological process* term GO:0008150) for each community.

The network inferred from Alzheimer/Control *H. sapiens* RNA-seq dataset exhibits a high sparsity level (≥99.14%), and a high modularity (≥0.67). The characterization of the GXN•OMP model obtained in terms of topological and inner-evaluation metrics are shown in Figure 12. The median number of regulators involved in the control of each gene was approximately equal to 12 regulators, while each regulator has 88 TGs in median. In terms of inner quality, the $R^{2}$ determination coefficients obtained for the prediction of TGs’ unseen expressions is lower than those obtained for the other RNA-seq datasets, but remain descent for a large population of genes, since the median $R^{2}$ score was close to 0.57 (only TGs with a score higher than 0.5 where kept for further analysis). Finally, the runtimes where comparable to those obtained for other RNA-seq datasets (0.076 seconds to fit TG’s model in median).

The Clauset-Newman-Moore technique has detected 40 communities. Among these communities, 26 exhibit significant GSEA Normalized Enrichment Scores and 18 exhibited enriched GO terms. Interestingly, 12 communities exhibited simultaneous GSEA and GO enrichment scores, suggesting that such communities may have important differential functional roles in Alzheimer disease and control groups respectively, as depicted in Figure 13. Some of these communities have many over-represented GO terms, which makes them particularly interesting to focus on, in a first analysis. For instance, community Hs|12 exhibits an enrichment for 33 GO terms and is over-expressed in the control group while community Hs|13 exhibits an enrichment for 22 GO terms and is over-expressed in the Alzheimer disease group. Therefore, Hs|12 and Hs|13 are important regulatory communities characterizing the Control/Alzheimer differences.

The analysis of the GO terms associated to the community Hs|13 reveals that it contains genes mostly involved in Immune Response in general, including antigen processing and presentation, interferon alpha/beta/gamma production and signaling, detection and defense response to viral infections. While the analysis of GO terms associated to Hs|12 revels more general biological processes related for instance to translation, mRNA splicing, processing and stability, amino acid metabolic process, catabolic processes (ubiquitin-dependent protein degradation) and signal transduction involved in tissue homeostasis (Wnt pathway).

The most important regulatory links (with coefficients higher than 0.2 in absolute values) within communities Hs|12 and Hs|13 are reported in Figure 14. Interestingly, in both sub-GRNs, the number of TGs that each TF controls varies to an important degree, exhibiting a scale-free like topology. The analysis of the top-10 TFs controlling the highest number of TGs, has revealed that most of these TFs have been found associated to Alzheimer disease in previous studies [77,78,79,80,81,82,83,84,85,86,87,88,89,90,91,92,93,94,95], as reported in Table 4. The full lists of TFs belonging to both communities are reported in Table A1.

## 4. Discussion

In this work we have proposed two *generalization-aware self-expressiveness* GRN inference tools, called GXN•OMP and GXN•EN, based respectively on OMP and ElasticNet linear regression algorithms. The generalization capabilities of the regression methods were optimized using an inner cross-validation procedure, and then evaluated using an outer cross-validation step.

We assessed the performance of both methods using the 4 datasets from well-known DREAM5 benchmark [24], and we compared their results to those obtained by state-of-the-art methods, namely SVR and RF. The quality of the results obtained by the self-expressiveness methods presented in this paper are comparable to those obtained by state-of-the-art methods, regarding external quality metrics and generalization capabilities. In addition, unlike state-of-the-art methods, the self-expressive models exhibit high levels of sparsity, making them directly analyzable, without requiring the application of post-processing steps (e.g., select top-k links, apply a threshold) to focus only on the most interesting links.

We have also applied both methods to three complex eukaryotic multi-tissue RNA-seq datasets. The GRNs inferred by these methods revealed to exhibit a sparse and modular structure, and their inner communities of TFs showed statistically significant over/under-expression on specific tissues and cell types, as well as significant enrichment for some anatomical GO terms, which suggests that such communities may also drive important functional roles. Indeed, both methods have revealed, for instance, communities of TFs particularly related to the development of the heart in *C. familiaris*, and neural tissues in *C. familiaris*, *R. norvegicus* and *H. sapiens*. A deeper analysis of some of these communities, revealed the presence of important TFs known to be involved in the development or differentiation of a given tissue. This shows the potential of the GXN approach to identify tissue-specific regulatory communities (i.e., sub-regulatory GRNs). This information can help to better understand the regulatory mechanisms underlying a given tissue identity. In biomedical applications, this information can also help detecting regulatory communities that are active in different tissues simultaneously, in order to analyze, for instance, if a treatment targeting specific TFs is susceptible to affect different tissues simultaneously. This approach could also be used to explore model organisms used in zoological, phylogenetic or agronomic studies, and for which the tissue identity has not been established at a molecular level.

Moreover, given the importance of the control/disease (or control/treatment) experimental configuration, and its associated differential expression analysis, in this work we analyzed an Alzheimer/Control RNA-seq study case, in order to illustrate how the GXN methodology can be used to perform differential GRN analysis. This analysis revealed 12 communities presenting significant GO and GSEA enrichment scores, and thus are likely to have important differential functional roles in Alzheimer disease and control groups respectively. In addition, a more in-depth analysis of the regulatory links present in two of these communities has revealed the links between important TFs that have shown to be involved in the Alzheimer disease in previous works. In order to infer tissue or condition-specific gene networks, a promising research direction seem the adaptation of this new formalism to the analysis of single-cell RNA-seq datasets [96].

The experiments and the methods implementations are available online in a dedicated gitlab repository https://gitlab.com/bf2i/gxn (last updated on 20 February 2023) and the software can be installed as a Python library, from the Python official third-party software repository for Python https://pypi.org/project/GXN/ (last updated on 20 February 2023). In addition to the implementation of the GXN•OMP and GXN•EN methods and dedicated tutorials, the GXN package includes different visualization and data exportation functions that would allow the users to represent and extract the GXN models regulatory links. These sources of information could be valuable inputs for biologist users in order to better understand the regulatory processes involving specific genes of interest.

The results obtained in this work suggest that *generalization-aware self-expressive gene networks* inference is a novel and promising methodology to produce, assess and tune, simple, interpretable, and predictive GRN models.

## Figures and Tables

**Figure 1 biomolecules-13-00526-f001:**
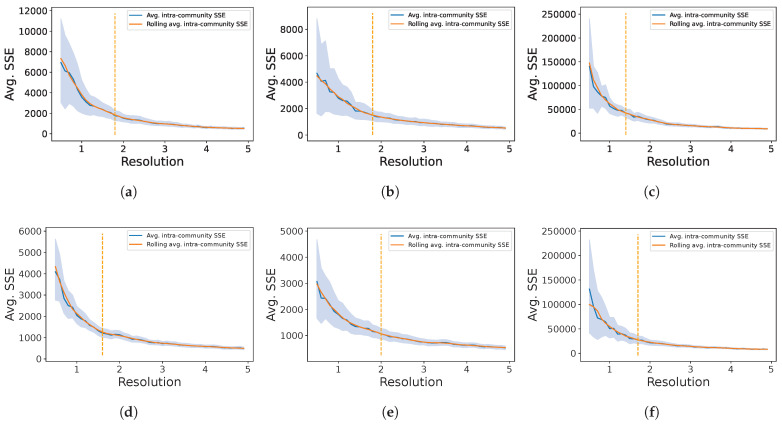
Average SSE as a function of the resolution parameter *r*. The elbow point (denoted as a vertical dashed line) was determined on the moving average plot (window = 5) in order to obtain more robust results. The retained retained resolutions were: r=1.8 (GXN•EN) and r=1.6 (GXN•OMP) for the *C. familiaris* dataset, r=1.8 (GXN•EN) and r=2 (GXN•OMP) for the *R. norvegicus* dataset, and finally r=1.4 (GXN•EN) and r=1.7 (GXN•OMP) for the *H. sapiens* dataset; (**a**) GXN•EN-*C. familiaris*; (**b**) GXN•EN-*R. norvegicus*; (**c**) GXN•EN-*H. sapiens*; (**d**) GXN•OMP-*C. familiaris*; (**e**) GXN•OMP-*R. norvegicus*; (**f**) GXN•OMP-*H. sapiens*.

**Figure 2 biomolecules-13-00526-f002:**
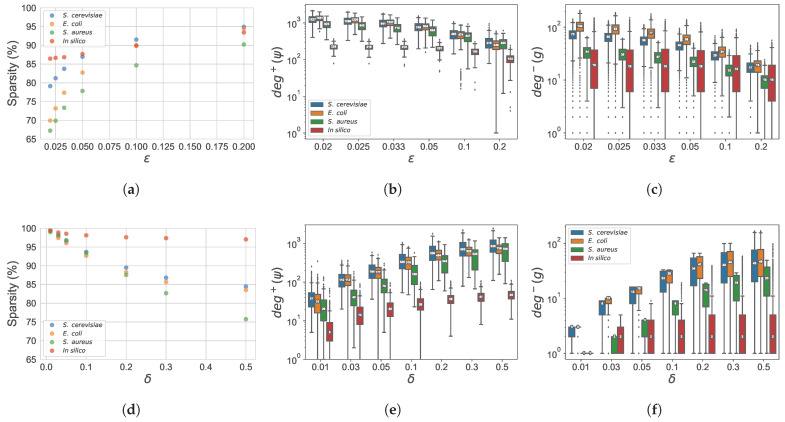
Sparsity of the networks inferred by the GXN•EN (**a**)) and the GXN•OMP (**d**)) methods. Distribution of regulators out-degree, for GXN•EN (**b**)) and GXN•OMP (**e**)). Distribution of target genes in-degrees for GXN•EN (**c**)) and GXN•OMP (**f**)). These results were obtained, on the DREAM5 dataset, with the different parameter settings presented in Section 2; (**a**) GXN•EN | sparsity; (**b**) GXN•EN | $deg^{+}\left(\psi \right)$; (**c**) GXN•EN | $deg^{-}\left(g\right)$; (**d**) GXN•OMP | sparsity; (**e**) GXN•OMP | $deg^{+}\left(\psi \right)$; (**f**) GXN•OMP | $deg^{-}\left(g\right)$.

**Figure 3 biomolecules-13-00526-f003:**
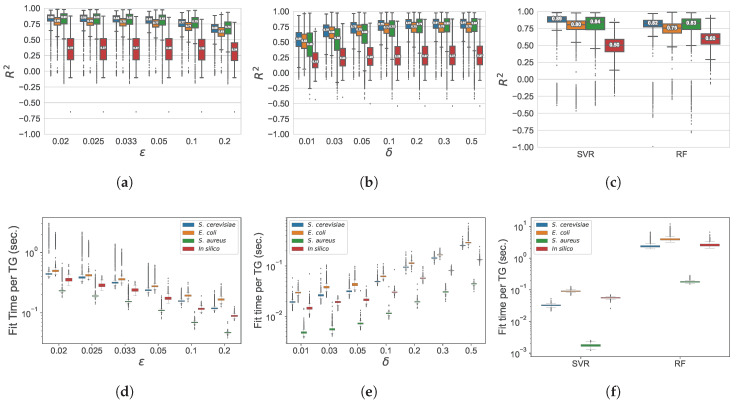
Average cross-validation $R^{2}$ scores for the GXN•EN (**a**), the GXN•OMP (**b**), as well as SVR and RF (**c**) state-of-the-art models. Average training run-times for the GXN•EN (**d**), the GXN•OMP (**e**), as well as SVR and RF (**f**) state-of-the-art methods. These results were obtained, on the DREAM5 dataset; (**a**) GXN•EN | $R^{2}$; (**b**) GXN•OMP | $R^{2}$; (**c**) GXN•EN | Run-time; (**d**) GXN•EN | Run-time; (**e**) GXN•OMP | Run-time; (**f**) SVR and RF | Run-time.

**Figure 4 biomolecules-13-00526-f004:**
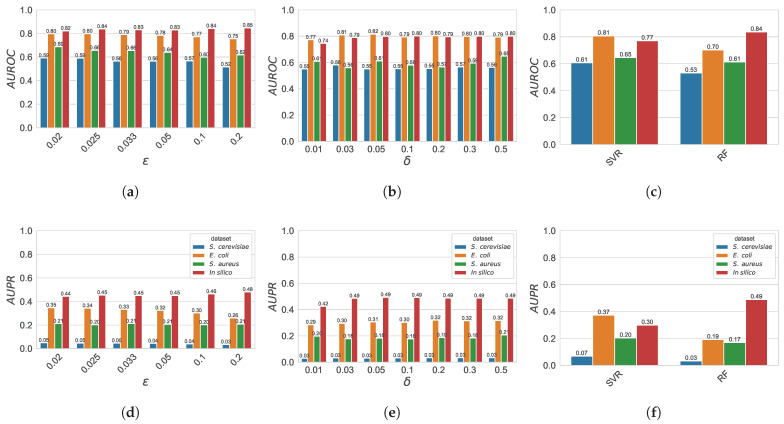
AUROC scores for the GXN•EN (**a**), the GXN•OMP (**b**), as well as SVR and RF (**c**) state-of-the-art models. AUPR scores for the GXN•EN (**d**), the GXN•OMP (**e**), as well as SVR and RF (**f**) state-of-the-art methods. These results were obtained, on the DREAM5 dataset; (**a**) GXN•EN | AUROC; (**b**) GXN•OMP | AUROC; (**c**) SVR and RF | AUROC; (**d**) GXN•EN | AUPR; (**e**) GXN•OMP | AUPR; (**f**) SVR and RF | AUPR.

**Figure 5 biomolecules-13-00526-f005:**
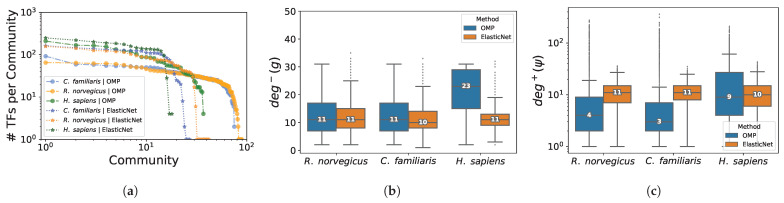
Number of TFs within each community (sorted in decreasing order) from the *C. familiaris*, *R. norvegicus* and *H. sapiens* networks inferred using the GXN•OMP and GXN•EN algorithms (**a**). Distribution of the target genes in-degree for the RNA-seq Eukaryote datasets (**b**) for the GXN•OMP and GXN•EN. Distribution of the regulators out-degree for the RNA-seq Eukaryote datasets (**c**) for GXN•OMP and GXN•EN; (**a**) Communities sizes; (**b**) Genes in-degree; (**c**) Regulators out-degrees.

**Figure 6 biomolecules-13-00526-f006:**
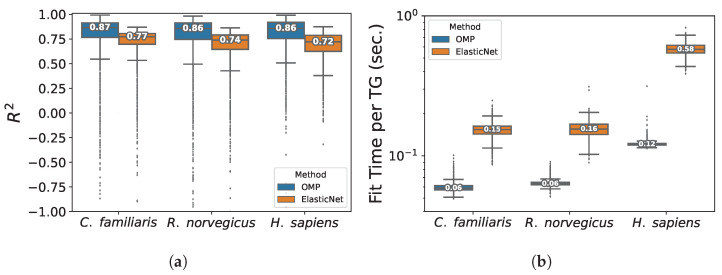
Average $R^{2}$ validation determination coefficients and average fitting run-times for the GXN•EN and the GXN•OMP algorithms in the RNA-seq eukaryotic datasets; (**a**) Mean $R^{2}$ determination coefficient; (**b**) Mean Fit time.

**Figure 7 biomolecules-13-00526-f007:**
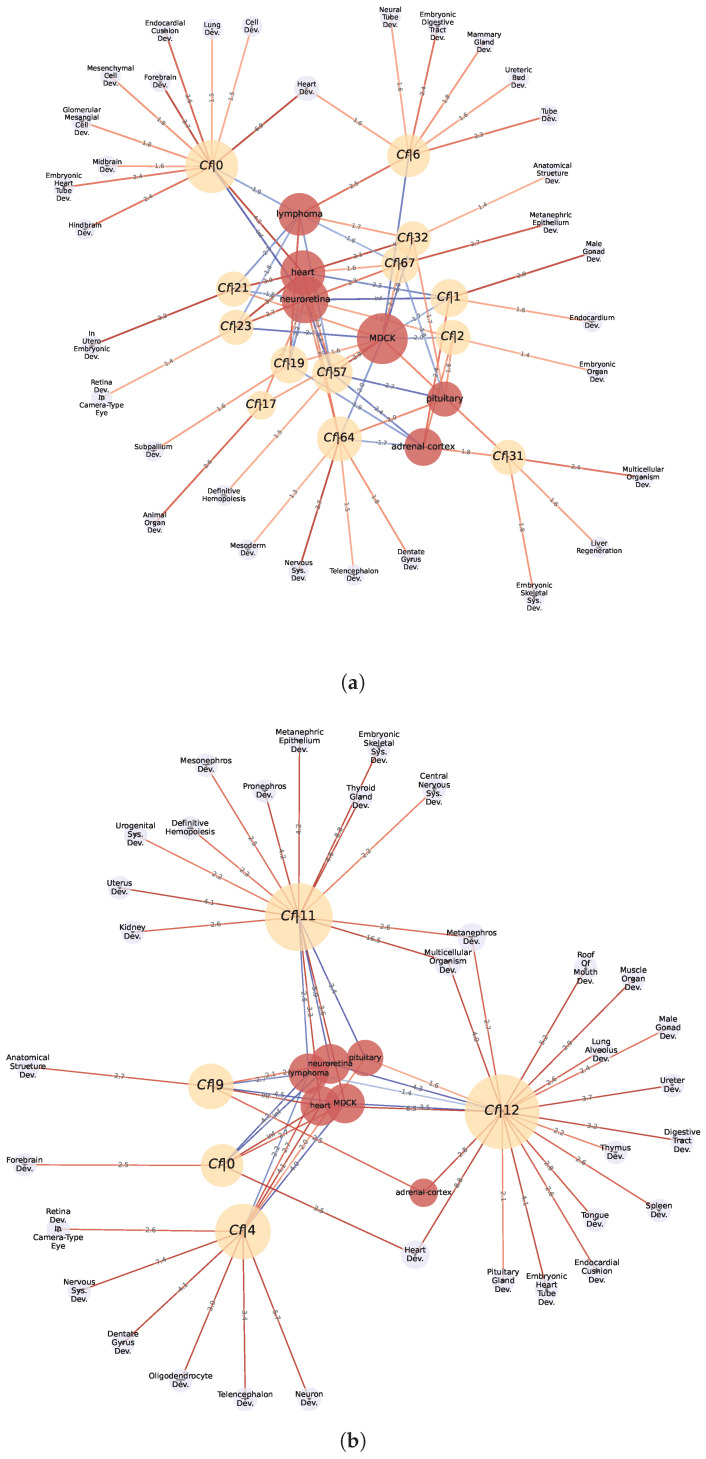
Statistically significant Normalized Enrichment GSEA Scores between cell-types (red nodes) and GXN•OMP and GXN•EN GRN communities (yellow nodes), and statistically significant associations between GO terms (blue nodes) and GXN•OMP and GXN•EN communities, for the *C. familiaris* dataset; (**a**) GXN•OMP-*C. familiaris*; (**b**) GXN•EN-*C. familiaris*.

**Figure 8 biomolecules-13-00526-f008:**
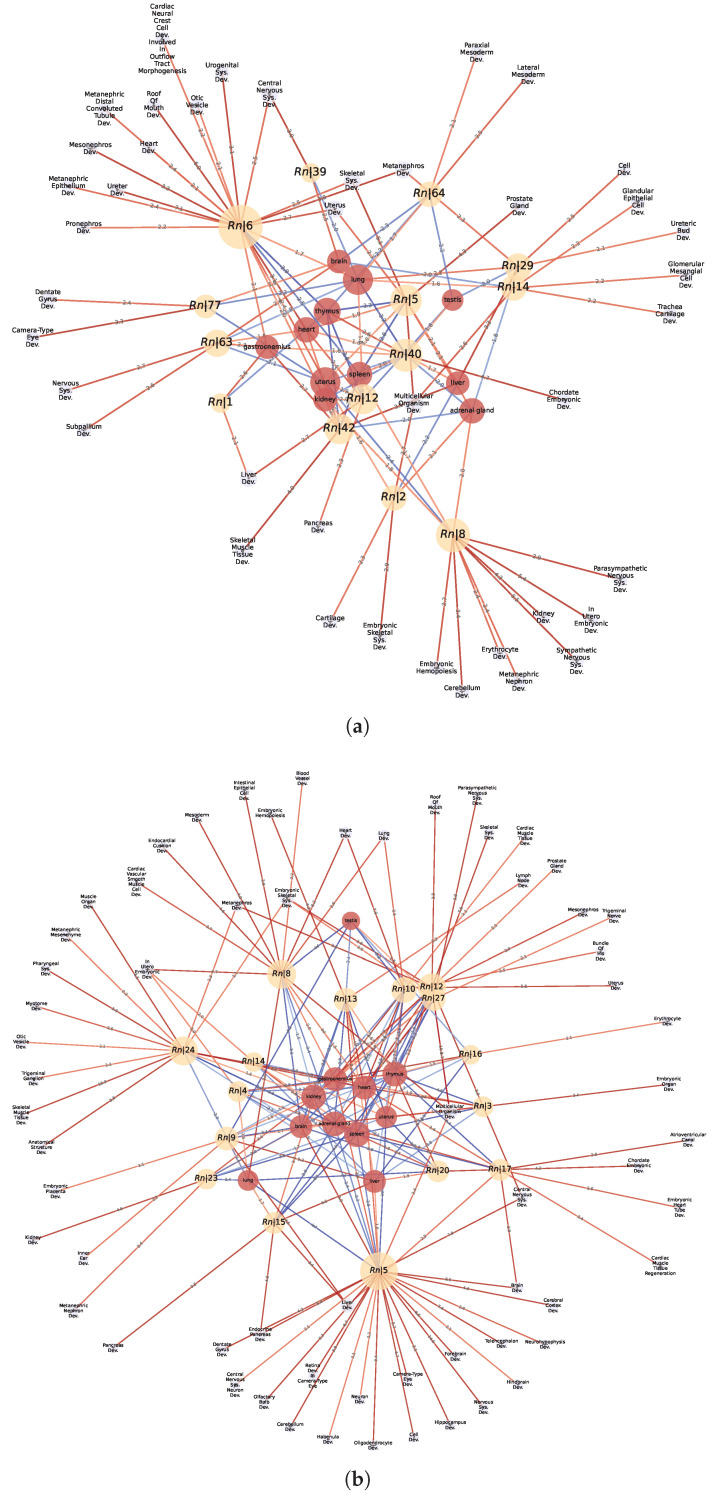
Statistically significant Normalized Enrichment GSEA Scores between cell-types (red nodes) and GXN•OMP and GXN•EN GRN communities (yellow nodes), and statistically significant associations between GO terms (blue nodes) and GXN•OMP and GXN•EN communities, for the *R. norvegicus* dataset; (**a**) GXN•OMP-*R. norvegicus*; (**b**) GXN•EN-*R. norvegicus*.

**Figure 9 biomolecules-13-00526-f009:**
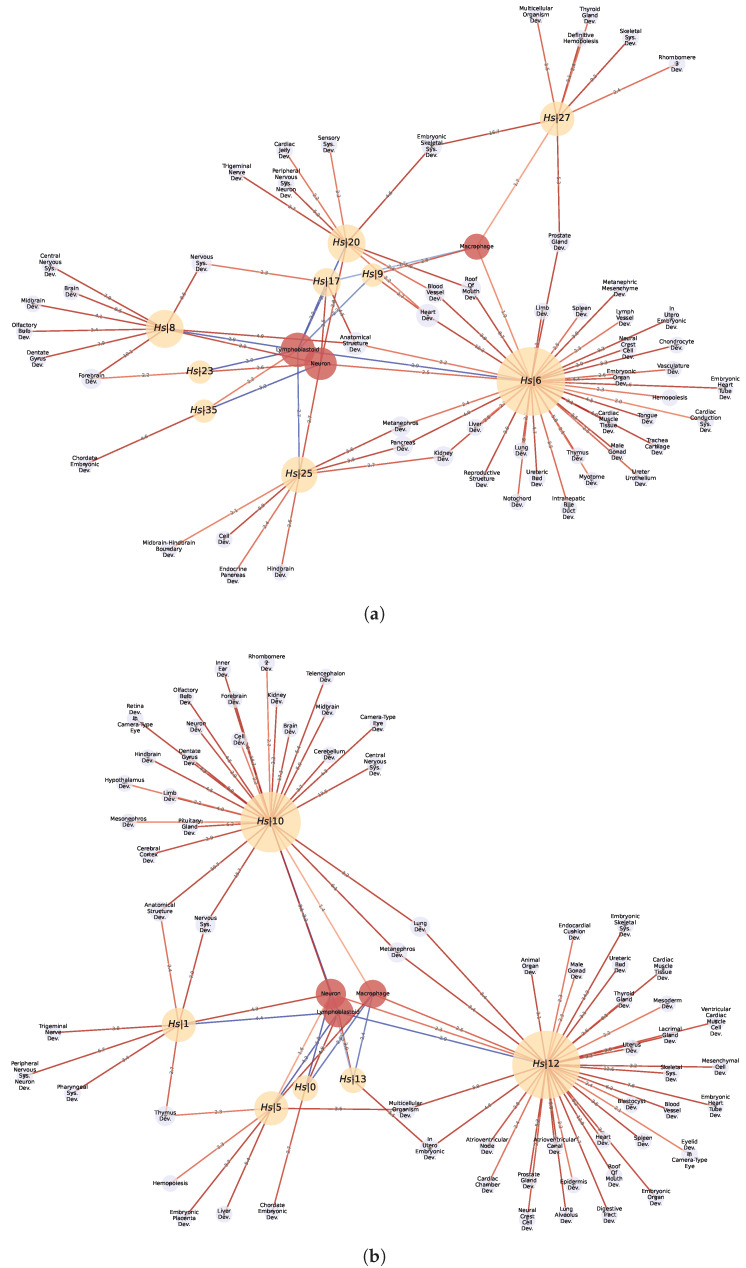
Statistically significant Normalized Enrichment GSEA Scores between cell-types (red nodes) and GXN•OMP and GXN•EN GRN communities (yellow nodes), and statistically significant associations between GO terms (blue nodes) and GXN•OMP and GXN•EN communities, for the *H. sapiens* dataset; (**a**) GXN•OMP-*H. sapiens*; (**b**) GXN•EN-*H. sapiens*.

**Figure 10 biomolecules-13-00526-f010:**
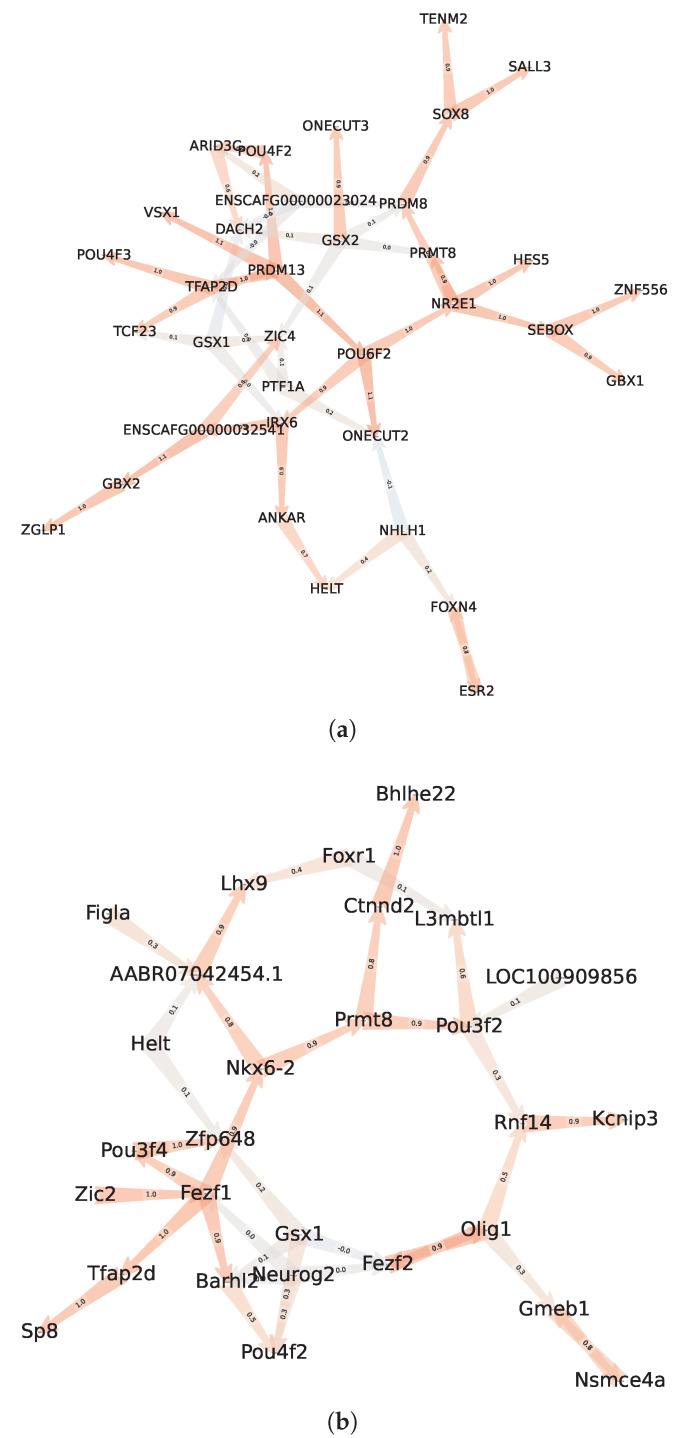
Regulatory communities Cf|23 (**a**) and Rn|39 (**b**) obtained applying GXN•OMP on the *C. familiaris* and *R. norvegicus* datasets respectively. The communities Cf|23 exhibits enriched GO terms related to neuroretina and retina development, while Rn|39 exhibits enriched GO terms related to brain and nervous system development; (**a**) Community *C. familiaris* GXN•OMP Cf|23; (**b**) Community *R. norvegicus* GXN•OMP Rn|39.

**Figure 11 biomolecules-13-00526-f011:**
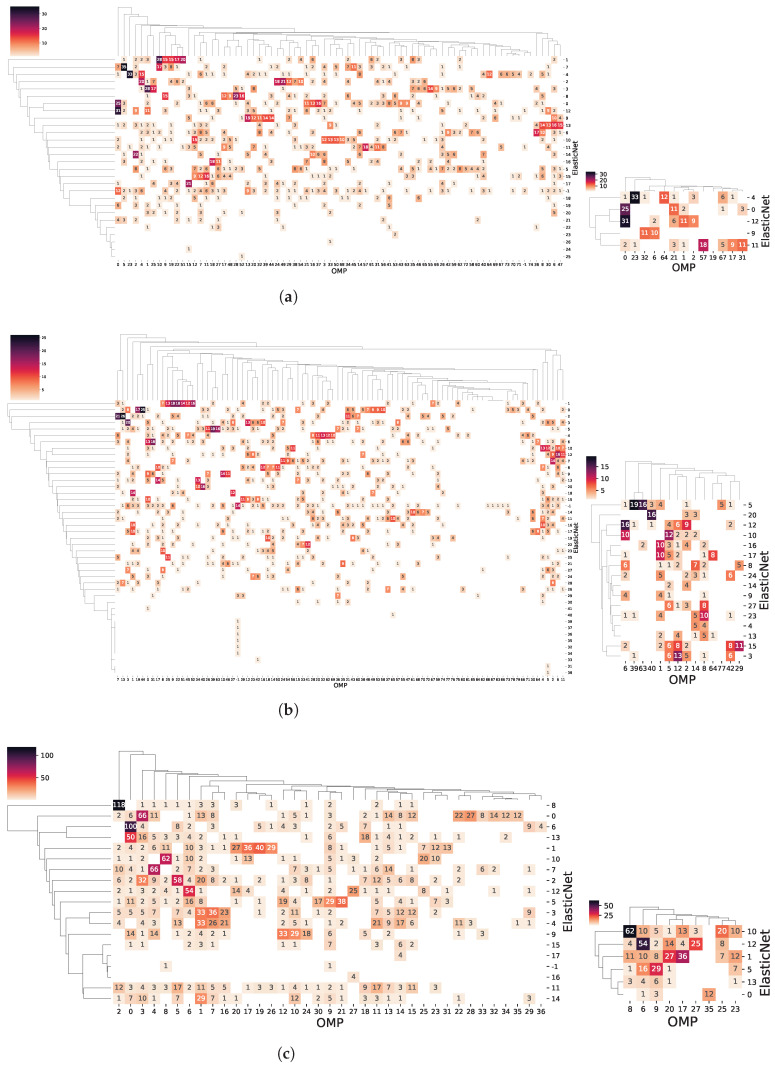
Confusion matrices between GXN•OMP and GXN•EN communities for the *C. familiaris*, *R. norvegicus* and *H. sapiens* datasets. Matrices on the left side represent the full comparison between GXN•OMP and GXN•EN communities, while matrices on the right side represent only the communities enriched in terms of GSEA and GO, depicted in (**a**–**c**); (**a**) *C. familiaris*; (**b**) *R. norvegicus*; (**c**) *H. sapiens*.

**Figure 12 biomolecules-13-00526-f012:**
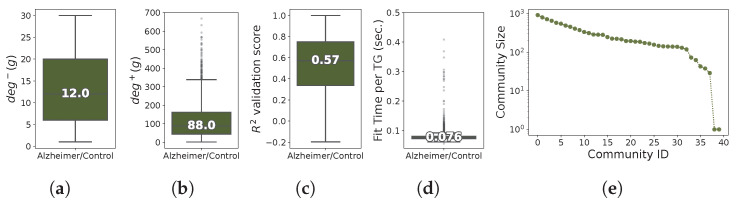
Distribution of TGs in-degree (**a**), TFs out-degree (**b**), $R^{2}$ determination coefficients (**c**), run-times (**d**) needed to train each TG model, and number of TGs within each community sorted in decreasing order (**e**), for the *H. sapiens* Alzheimer/Control dataset; (**a**) TGs in-degree; (**b**) TFs out-degrees; (**c**) $R^{2}$ score; (**d**) Run-time; (**e**) Communities sizes.

**Figure 13 biomolecules-13-00526-f013:**
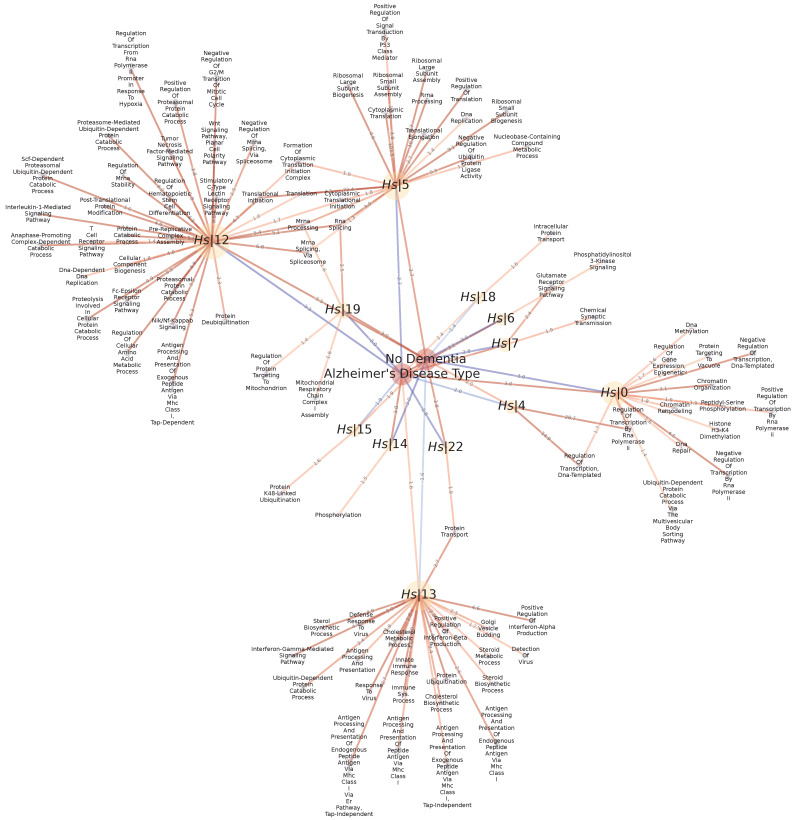
Statistically significant Normalized Enrichment GSEA Scores between Disease/Control conditions (**red nodes**) and GXN•OMP communities (**yellow nodes**), and statistically significant associations between GO terms (**blue nodes**) and GXN•OMP and GXN•EN communities, for the Alzheimer/Control *H. sapiens* dataset.

**Figure 14 biomolecules-13-00526-f014:**
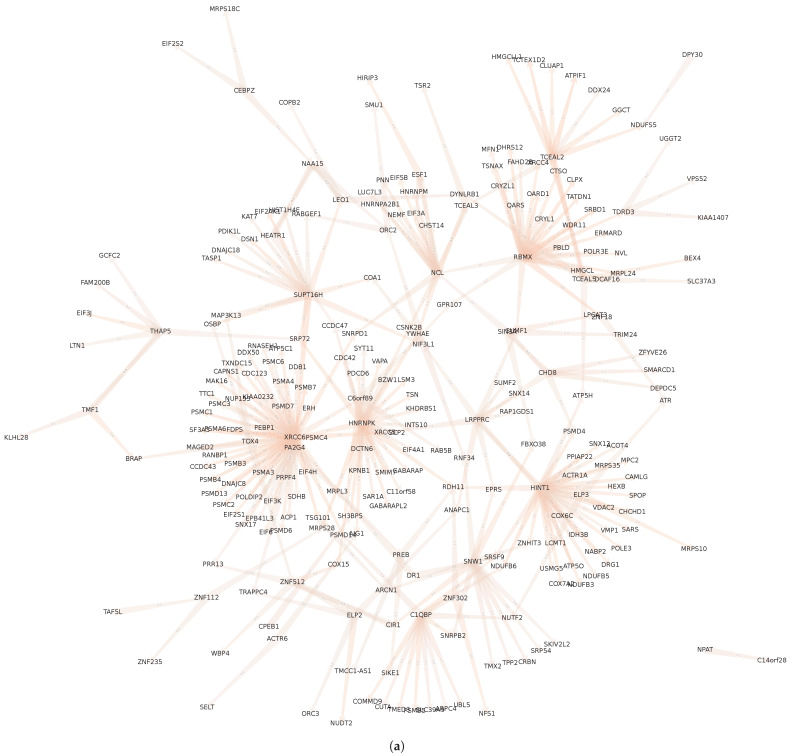
Regulatory communities Hs|12 and Hs|13, within the GXN•OMP obtained from the *H. sapiens* Alzheimer/Control dataset. Communities Hs|12 is over-expressed in the control samples and under-expressed in Alzheimer disease samples, while community Hs|13 depicts the opposite expression pattern. For the sake of clarity, we represent only the links with a coefficient higher than 0.2 in absolute values; (**a**) Community Hs|12; (**b**) Community Hs|13.

**Table 1 biomolecules-13-00526-t001:** DREAM5 Benchmark and RNA-seq eukaryotic dataset summary.

DREAM5	Data	*D*	|Γ|	|Ψ|	$|E_{\mathrm{gold}}^{}|$	$\frac{|E_{\mathrm{gold}}^{}|}{|E_{\mathrm{gold}}^{\mathrm{full}}|}$	$|E^{\mathrm{full}}|$
in silico	Simulated	805	1643	195	4012	0.014	320,190
*S. aureus*	Microarray	160	2810	99	515	0.028	278,091
*E. coli*	Microarray	805	4511	334	2066	0.013	1,506,340
*S. cerevisiae*	Microarray	536	5950	333	3940	0.017	1,981,017
Eukaryotes	Data	*D*	# Tissues	|Ψ| = |Γ|	$|E^{full}|$		
*C. familiaris*	RNA-seq	75	6	2286	5,223,510		
*R. norvegicus*	RNA-seq	80	11	2358	5,557,806		
*H. sapiens*	RNA-seq	657	3	2454	6,019,662		
Control/Disease	Data	*D*	|Γ|	|Ψ|	$|E^{full}|$		
*H. sapiens*—Brain Control/Alzheimer	RNA-seq	377	17,574	1994	35,042,556		

**Table 2 biomolecules-13-00526-t002:** Sparsity level for each DREAM5 dataset, and each algorithm, GXN•OMP and GXN•EN models exhibit high sparsity levels unlike SVR and RF state-of-the-art models.

	*S. cerevisiae*	*E. coli*	*S. aureus*	In Silico
SVR	0.025%	0.027%	0.021%	0.021%
RF	0.036%	0.002%	0.002%	0.019%
GXN•OMP (Min)	84.474%	83.530%	75.760%	97.037%
GXN•OMP (Max)	99.258%	99.181%	98.990%	99.487%
GXN•EN (Min)	79.119%	69.949%	67.240%	86.456%
GXN•EN (Max)	94.929%	94.454%	90.238%	93.429%

**Table 3 biomolecules-13-00526-t003:** RNA-seq Eukaryotic datasets sparsity and modularity.

	Sparsity (%)		Modularity	
	GXN•EN	GXN•OMP	GXN•EN	GXN•OMP
*R. norvegicus*	99.495	99.39	0.575	0.829
*C. familiaris*	99.491	99.357	0.627	0.835
*H. sapiens*	99.538	99.087	0.658	0.573

**Table 4 biomolecules-13-00526-t004:** Out-degree and possible relations to Alzheimer disease in the literature, for the top-10 TFs from communities Hs|12 (over-expressed in control and under-expressed in Alzheimer) and Hs|13 (under-expressed in control and over-expressed in Alzheimer).

Community Hs|12 Alzheimer (−) Control (+)			Community Hs|13 Alzheimer (+) Control (−)		
**TF** **(** ψ **)**	$\deg^{+}\left(\psi \right)$	**Alzheimer** **Link**	**TF** **(** ψ **)**	$\deg^{+}\left(\psi \right)$	**Alzheimer Link**
XRCC6	52	[77]	YWHAB	63	[78,79]
HINT1	34	[80]	BTN3A3	31	[81]
RBMX	30	-	PARP14	21	[82]
HNRNPK	30	[83]	TRIM25	20	[84]
CPEB1	27	[85]	STAT1	20	[86]
NCL	18	[87]	CGGBP1	19	[88]
SRA1	17	[89]	SREBF2	18	[90]
SUPT16H	17	-	PARP9	17	[91]
ACTR6	15	[92]	ZBTB38	17	[93]
SNW1	15	[94]	PICALM	17	[95]

## Data Availability

The scripts and the methods implementations as well as the *C. familiaris*, *H. sapiens*, *R. norvegicus* and the Alzheimer vs. Control datasets are available online at https://gitlab.com/bf2i/gxn (last updated on 20 February 2023). And the DREAM5 dataset is available at https://www.synapse.org/#!Synapse:syn2787209/wiki/70350 (last accessed 20 February 2023). The software can be installed as a Python library, from the Python official third-party software repository for Python https://pypi.org/project/GXN/ (last updated on 20 February 2023).

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
