# Peer review of "Gene Self-Expressive Networks as a Generalization-Aware Tool to Model Gene Regulatory Networks"

_biomolecules, 2023, doi:10.3390/biom13030526_

Round 1
Reviewer 1 Report
It is not clear that the generalization capacity of a model is something different from overfitting, when the authors justify how their use of the self-expression network models. Also in GRN literature, the most typical metric for evaluation is by comparing the edges in the graph with those in the ground truth via ROC or precision-recall curves. The R^2 or residual errors are rarely used for GRN performance evaluation, as some function may predict the values correctly by complex functions but not necessarily capturing the network structure. The results also lack biological support regarding the links found in the inferred networks. It is not clear how a biologist user may benefit from the results.
Reviewer 2 Report
Gene Self-Expressive Networks as a generalization-aware tool to model gene regulatory networks
The author has presented a new tool for building a gene regulatory network (GRN) using a self-expressiveness model. They have trained their model using two methods Orthogonal Matching Pursuit (OMP) and Elastic Net (EN) and evaluated on different RNA or microarray datasets of different organisms using cross validation. The model performance has good R2, AUROC, and AUPR value. The output of these models has over / under expression of genes with significant enrichment of GO terms. I think the gene self-expressive networks could be useful for GRN using the RNA seq dataset. However I would suggest that author should make the following changes regarding the interpretation/analysis of the data:
Major Correction:
-
If the model is using different experimental conditions, does it also take care of the batch effect in the gene expression value?
-
The assumption of the Generalization gene self expression relies on the self regulatory transcription factor (TF) that regulates the gene in the same subspace and controls its own expression i.e. self expression. However in the real world it is not like that. It is not compulsory that all transcription factors control their own expression. In most of the cases this is not the scenario. How does the author deal with this issue?
-
The self expression of TF had an effect on the whole GXN models. It raises the question that the factors that are regulating the TF not regulating by its own expression won't be identified. This leads to the problem that the model won't be able to provide the whole picture of the GRN. How does the author deal with this issue?
-
Since epigenetic changes are quite often in the genome. Even at each cell type level as well as their differentiation or activation stage the epigenetic of the cell changes quite frequently. Then how does the author explain the proxy GRN (RegPrecise) for S. aureus could be considered as a gold standard?
-
Author should explain in more detail how Ordinary Differential Equations (ODE) are used in generating the gene expression matrix from the GRN. What model and dynamics are being used? Please elaborate. Page 7 line 295-297.
-
Line 330 to 332, “Links reported in the gold standards, are taken as true interactions, while only pairs missing from the gold standard and involving both a TF and a TG experimentally studied, are taken as false interactions. ” Why is the experimental interaction considered a false interaction? What does this True False means, does it involve upregulation or downregulation of the TG or TF?
-
In Figure 7, the author has shown the GSEA outcome of the communities with the gene ontology. However I would expect the network with the transcription factor that are regulating the genes. Also TFs are regulating each other TFs that form gene regulatory networks. The GRN would provide the idea to the user to target which gene or TFs so the reader can work on that gene or TF. Can the author build a GRN with the TFs and target genes in different datasets and give interpretations?
-
In the real world, the GRN is performed between control and treatment. Can the author please use the dataset of the disease and control and make the GRN and show how different they are?
-
Author has not properly presented their dataset interpretation in the manuscript. I think Figure 7,8, and 9 has not been even mentioned in the manuscript. That is one of the major parts of the manuscript that determines how the reader would evaluate or interpret their data once obtained from the GXN-OMP or GXN-EN model. Can the author please interpret their data in more detail and also explain the figure 7,8, and 9 in detail?
-
The discussion part is not well explained.I advise the author to provide their discussion in more detail.
-
The author has focused more on the methodology and introduction rather than results and discussion. I suggest to the author that both the result and discussion should be more detailed and have more interpretation of the data.
-
I could not find the github page so that I can use this model for my dataset. Can the author provide the trained model for different organisms that can be used by others for their dataset interpretations or building GRN?
Minor Correction:
-
Page 1 line 23 instead of “regulator genes” change to “regulatory module”.
-
Please give some examples of Probabilistic Model-Based methods as did for Multi-Network methods SCENIC.
-
Page 7 Line 299 correct G inference to GRN inference.
-
Line 476 and 477, please use the full form of SVR and RF in order to avoid any confusion to the reader.
-
Line 488 and other places in the manuscript where you have mentioned seconds. It is not for the full model but per TG. Please mention it like that.
-
In line 545, The figure number has not been mentioned, instead there are two question marks (??).
Round 2
Reviewer 2 Report
The authors has successfully addressed the comments in details.